# Presentations to the Emergency Department for Problems Related to Mental Health: Sex Differences in Adolescents

**DOI:** 10.3390/ijerph192013196

**Published:** 2022-10-13

**Authors:** Raffaela M. Flury, Lara Brockhus, Martin Müller, Jonathan Henssler, Aristomenis K. Exadaktylos, Jolanta Klukowska-Rötzler

**Affiliations:** 1Department of Emergency Medicine, Inselspital, University Hospital Bern, University of Bern, CH-3010 Bern, Switzerland; 2Department of Psychiatry and Psychotherapy, University of Cologne Medical School, DE-50937 Cologne, Germany; 3Charité University Medicine, St Hedwig-Krankenhaus, Clinic for Psychiatry and Psychotherapy, DE-10117 Berlin, Germany

**Keywords:** mental health problems, emergency department, adolescents

## Abstract

Background: Adolescents aged sixteen to eighteen years are mostly treated in adult emergency departments. In a study at our tertiary adult emergency department (ED) at the University Hospital in Bern (Inselspital), Switzerland, we found that adolescents significantly more often present with psychiatric problems than adults. The study at hand aimed to characterise those adolescents presenting to the ED triaged with a chief complaint regarding mental health. Furthermore, the goal was to assess sex differences in terms of diagnosis, suicidal ideation, substance use, as well as social impact. Methods: We conducted a single-centre, retrospective review of presentations to our adult ED related to the mental health of adolescents aged 16 to 18 years, covering the period from January 2013 to July 2017. Anonymised data were extracted from medical reports referring to the ED visits that were triaged as mental-health-related, and we assessed these for diagnosis, acute and previous suicidal ideation, history of self-harm, external aggression, substance use and social problems. We focused on patient characterisation and defining sex differences. Results: Data were analysed for a total of 612 consultations by adolescents who presented to our ED with problems related to mental health. Women accounted for 61.1% of cases. The most frequent diagnoses were reactions to severe stress and adjustment disorders (19.1%), followed by alcohol use disorders (17.6%), intentional self-harm (17.3%), and affective disorders (13.7%). Males had lower odds for intentional self-harm (OR 0.10, 95% 0.05–0.21, *p* < 0.001) as well as disorders of personality and behaviour (OR 0.09, 95% 0.21–0.37, *p* < 0.001), whereas they had higher odds of being admitted due to use of alcohol (OR 2.51, 95% 1.65–3.83, *p* < 0.001). Of all cases, 31.7% reported acute suicidal ideation, with a significantly lower odds ratio in males (OR 0.58, 95% 0.41–0.84, *p* = 0.004). The most common source for referral to the ED was family members (25.2%). Males had twice the odds of reporting alcohol consumption as well as use of cannabis (in both cases *p* < 0.001). In 27.9% of all cases, familial problems were reported, with males having significantly lower odds of being exposed to these (OR 0.64, 95% 0.44–0.94, *p* = 0.021), whereas they had higher odds of reporting educational problems (OR 1.68, 95% 1.04–2.72, *p* = 0.035). Conclusions: Adolescents aged sixteen to eighteen years presenting to the ED with problems related to mental health are an important subgroup of ED presentations and should be thoroughly assessed for suicidal ideation, substance use, as well as familial and educational problems. Assessment and establishment of post-ED care are of particular importance in this vulnerable patient group.

## 1. Introduction

At the University Hospital in Bern (Inselspital), Switzerland, patients aged 16 years and older are treated at the adult emergency department (ED), whereas younger patients are referred to paediatric emergency services. Therefore, adolescents aged sixteen to eighteen years are a transitional age group, interposed between paediatric and adult ED services. In a study at our tertiary ED at the Inselspital, we have shown that these adolescents present significantly more often with trauma and psychiatric problems than adults [1]. The study at hand aimed to gather more detailed information about the adolescents aged 16 to 18 years presenting with psychiatric problems to an adult Swiss ED, in order to characterise this particular group of patients more clearly.

It is known that mental health disorders in children and adolescents are becoming more frequent worldwide, particularly in developed countries [2]. Lo et al. showed that over a ten-year study period (2007 to 2016) in the United States, paediatric ED visits due to mental health disorders rose by 60%, whereas the overall paediatric ED visits were stable [3]. The greatest increase (of 68%) was observed among adolescents aged 15 to 18 years. According to the World Health Organization (WHO), approximately 10% to 20% of adolescents globally experience mental health problems, although a considerable number remain underdiagnosed and undertreated [4]. In addition, they stated that 16% of the global burden of disease in teenagers aged 10–19 years is due to mental health conditions.

In Switzerland, three of the top 10 causes of death in adolescents aged 15–19 years are associated with mental health problems, namely self-harm, drug use disorders and interpersonal violence [5]. If the top ten causes of disability-adjusted life years (DALYs) are scrutinised, the importance of mental illness in adolescents becomes even clearer: six of the top 10 causes of DALYs for Swiss adolescents aged 15–19 years are psychiatric problems, with anxiety and depressive disorders being the top two [6]. Further mental-health related problems causing the loss of healthy life-years are self-harm, drug use disorders, childhood behavioural disorders and bipolar disorder.

Adolescence is a vulnerable phase, in which many biological, cognitive, emotional, and social changes occur [7]. Essential developmental tasks include shaping one’s identity, detachment from home and obtaining independence, career entry as well as building one’s social network. As many mental disorders emerge during adolescence, they continue to have substantial effects on life into adulthood [8]. It is known that mental disorders are associated with educational underachievement, substance abuse, violence, and increased health-care costs. In addition to this, they pose a major risk factor for self-harm and suicide, which is the most common cause of death in Swiss adolescents aged 15 to 19 years [5,9]. The ages of 15–18 years mark a period of transition, including changes in health care systems. As mentioned before, in our ED, patients from the age of 16 years are treated by the adult emergency services. However, for adolescents up to 18 years, psychiatric care is guaranteed by child psychiatrists. The transitional phase of adolescents consequently poses a central challenge in psychiatric care [7].

As important and challenging as mental health in young adults is, few data exist on the characteristics of adolescent ED users presenting with psychiatric problems. The study at hand will address important questions such as: What are the differences among adolescent ED users in terms of current and past psychiatric diagnoses, social setting and use of drugs and alcohol? Furthermore, the goal was to distinguish sex differences in presentation with problems related to mental health, since a previous study on the whole Swiss population suggested that males more often present with externalised diseases such as hyperkinetic disorders, substance abuse, and antisocial behaviour, whereas females more frequently suffer from depressive and anxiety disorders, as well as eating and psychosomatic disorders [9]. This unique retrospective study of young ED users will fill a gap in our current knowledge of the patterns of psychiatric problems suffered by adolescents utilising our University Emergency Department. The data may lay the foundation for further inquiries and help to guide public health measures.

## 2. Materials and Methods

### 2.1. Setting and Study Population

We conducted a single-centre, retrospective review of mental-health-related presentations of adolescents aged 16 to 18 years to our tertiary adult emergency department (ED) at the University Hospital in Bern (Inselspital), based on anonymised patient data in the period of January 2013 to July 2017.

### 2.2. Data Collection and Extraction

The study included patients from the age of 16 up to their eighteenth birthday who were triaged as presenting to the ED for mental health problems. The data for this study were generated from the database of the management system of the University Hospital in Bern, Switzerland (Ecare, Turnhout, Belgium). Only medical reports referring to the ED visit were taken into consideration.

At our ED, patients are routinely triaged using the Swiss Emergency Triage Scale (SETS), an abbreviated version of the Manchester Triage System. Specially trained nurses work with a defined algorithm that considers vital signs to categorise patients according to their reported complaints and treatment priority. For the study at hand, patients who were triaged as having mental-health-related problems were selected.

Exclusion criteria were errors at triage, meaning that the chief complaint was not a psychiatric problem: these included unintentional ingestion of a foreign body or body packing (triaged as “foreign body”), smoke inhalation or chemical burns (triaged as “overdose”), as well as mental confusion due to infections, epileptic seizures, migraine, or syncope (triaged as “psychiatric problems”). In addition, patients with incomplete medical reports and therefore missing data (missing diagnosis and/or missing documentation of medical assessment) were excluded from the analysis.

Based on their final diagnosis according to ICD-10, patients were grouped into 15 categories [10]:Mental and behavioural disorders due to use of alcohol (F10)Mental and behavioural disorders due to other psychoactive substances (F11–F18)Mental and behavioural disorders due to multiple drug use (F19)Schizophrenia, schizotypal and delusional disorders (F20–F29)Affective disorders (F30–F39)Anxiety disorders, and obsessive compulsive disorders (F41–F42)Reaction to severe stress, and adjustment disorders (F43)Dissociative and somatoform disorders (F44–F45)Behavioural syndromes associated with physiological disturbances and physical factors (F50–F59)Disorders of adult personality and behaviour (F60–F69)Disorders of psychological development (F80–F89)Hyperkinetic disorders (F90)Symptoms and signs involving emotional state (R45)Suicidal tendency (R45.8), meaning triaged as presentation to the ED due to an attempted suicideIntentional self-harm (X84)

The following parameters were examined from medical records referring to the ED visit: sex, diagnosis, migration background, source for admission, stay after discharge, previous psychiatric care, acute and previous suicidal ideation, external aggression, history of self-harm, history of alcohol use as well as smoking and drug use, history of traumatic events and social problems. It is emphasised that there were no standardised questionnaires, but information was gathered from assessment of medical history and examination by the doctor on duty. In some cases, patients were additionally seen by a psychiatrist. We exclusively sampled anonymised visits, so that multiple visits by the same patient may have been included separately.

### 2.3. Ethical Considerations

This study was approved by the cantonal (district) ethics committee in Bern (2020-01388) and followed the guidelines of the Declaration of Helsinki and ethical principles for conducting medical research with human subjects. No individual informed consent was obtained. All data were anonymised prior to analysis.

### 2.4. Statistical Analyses

Data was summarised using descriptive statistics (mean values, percentages). The statistical analysis was performed using Stata^®^ 16.1 (StataCorp, The College Station, TX, USA). The distribution of categorical variables is given with the absolute number and the relative number as a percentage. The comparison of categorical variables was analysed using the chi squared test (χ2). The threshold of significance was set at *p* = 0.05 (two tailed). A total of 60 hypotheses were tested. Odds ratios (ORs) with 95% confidence intervals (CI) for all parameters were calculated with univariable logistic regression.

## 3. Results

### 3.1. Study Population

During the period under review, 6559 adolescents (as defined above) were treated at our adult ED. A total of 689 ED visits were triaged as mental-health-related diagnoses. A total of 24 cases (3.5%) wrongly triaged as mental-health-related were excluded after the assessment of medical records. In the end, a total of 665 cases presented to the ED with actual mental-health-related problems, corresponding to 10.1% of all cases among adolescents during the study period (Figure 1).

Another 53 patients were excluded due to missing data in the medical reports, with a final total of 612 cases included in the statistical analysis. A total of 61.1% of all cases were female patients (n = 374), compared to 38.9% male patients (n = 238), corresponding to a ratio of 1.6:1 (female: male).

### 3.2. Diagnosis

As seen in Table 1 and Figure 2, adolescents with mental-health-related problems most frequently presented with reactions to severe stress and adjustment disorders (n = 117, 19.1%), followed by disorders due to use of alcohol (n = 108, 17.6%), intentional self-harm (n = 106, 17.3%) and affective disorders (n = 84, 13.7%). Male patients were significantly less likely to present for intentional self-harm (OR 0.10, 95% 0.05–0.21, *p* < 0.001), as well as disorders of personality and behaviour (OR 0.09, 95% 0.21–0.37, *p* < 0.001). In contrast, significantly more males presented with disorders due to use of alcohol (OR 2.51, 95% 1.65–3.83, *p* < 0.001). In addition, males had higher odds for disorders of psychological development, although the overall low incidence led to a wide confidence interval (OR 11.3, 95% 1.38–92.46, *p* = 0.024).

### 3.3. Suicidal Ideation and History of Previous Psychiatric Assistance

Suicidal tendencies as the main reason for presenting to the ED and therefore set as primary diagnosis were only identified in 1.1% (n = 7) of all patients; however, as Table 2 and Figure 3 show, in almost a third of the cases (n = 194, 31.7%) acute suicidal ideation was reported on the explicit assessment at the ED, with significantly lower odds in male patients than in females (OR 0.58, 95% 0.41–0.84, *p* = 0.004). Males were three times more likely than females to show aggressive behaviour towards others (OR 3.11, 95% 1.73–5.60, *p* < 0.001). A total of 54% of our cohort had sought psychiatric assistance before their presentation to emergency services, with males having significantly lower odds for this (OR 0.46, 95% 0.33–0.64, *p* < 0.001). In addition, males were significantly less likely to report previous suicidal intentions (OR 0.44, 95% 0.32–0.61, *p* < 0.001), as well as previous suicide attempts (OR 0.27, 95% 0.16–0.44, *p* < 0.001) and self-harming behaviour (OR 0.25, 95% 0.16–0.39, *p* < 0.001) (Table 2, Figure 3*)*. Overall, one fifth of all cases reported a prior suicide attempt and almost half of the cohort presented with prior intentions for suicide. In total, 2.5% of patients reported abuse as children, but without any significant sex differences.

Table 3 shows that in 41.7% of the cases, patients had already been diagnosed with a mental health problem prior to ED presentation. The largest group of these patients presented with a pre-existing disorder of adult personality and behaviour (n = 56, 9.2%), with males showing significantly lower odds than females (OR 0.17, 95% 0.07–0.40, *p* < 0.001). In addition, males were less likely to present with prior known affective disorders, whereas they had higher odds for hyperkinetic and conduct disorders.

### 3.4. Source for Admission

The most common source for referral to the ED were family members (n = 154, 25.2%), followed by peers (n = 94, 15.4%) and health care providers (n = 90, 14.7%) (Table 4). Males were almost twice as likely to be delivered to the ED by the police (OR 1.96, 95% 1.08–3.55, *p* = 0.026), whereas they were significantly less often admitted by health care professionals (OR 0.40, 95% 0.24–0.67, *p* < 0.001).

### 3.5. Substance Use

Table 5 shows that we found significant differences between female and male patients regarding substance abuse. In 43.3% (n = 265) of all cases, patients reported general alcohol consumption when asked about this (independent of a possible diagnosis of alcohol use disorder), with males being twice as likely to drink alcohol (OR 2.07, 95% 1.49–2.88, *p* < 0.001). Furthermore, they were twice as likely to use cannabis (OR 2.03, 95% 1.41–2.93, *p* < 0.001) and they were more likely to consume LSD (OR 3.28, 95% 1.30–8.24, *p* = 0.012). Overall, our results show that in approximately one quarter of the cases, adolescents had used cannabis before. Additionally, more than one fifth in both males and females reported smoking, but there was no significant sex difference.

### 3.6. Social Impact

Another important influencing factor was the social situation (Table 6), with 27.9% (n = 171) of all cases presenting with familial problems. With an OR of 0.64 (95% 0.44–0.94, *p* = 0.021), males were less often exposed to problems in the familial setting, whereas they were significantly more likely to encounter difficulties either in school or in their apprenticeship (OR 1.68, 95% 1.04–2.72, *p* = 0.035). A small number reported problems with peers and exposure to death, disease, or trauma of other persons, but we could not find significant differences between sexes for these factors. In our records, in approximately 14% of all cases, patients had migration background, with 2.58 higher odds for males (95% 1.61–4.11, *p* < 0.001).

### 3.7. Route of Discharge

More than half of the patients were released home (Table 7). Male patients were significantly more likely to be released home than females (OR 1.44, 95% 1.04–2.00, *p* = 0.028), but less likely to be admitted to the hospital or to an external psychiatric clinic after ED presentation. However, this difference did not reach significance.

Adolescents admitted to a psychiatric ward are summarised under “external hospital”, since all the psychiatric clinics are outside of our University Hospital. Hospital admittance only occurred for patients needing to be treated in the ICU or for general medical conditions or surgical treatment.

## 4. Discussion

To our knowledge, this is the first Swiss study to characterise adolescents aged 16 to 18 years presenting to general adult emergency services due to mental health problems. Overall, one tenth of cases of adolescents presenting to our ED were triaged as having a psychiatric problem [1]. This shows the importance of understanding mental health conditions among adolescents, especially since these years are crucial in development and maintenance of social and emotional behaviour, which are in turn important for mental well-being [4]. Furthermore, there are few studies from any country that focus on this particular transitional age group. Our present study at hand presents a good clinical characterisation of a subgroup of patients presenting to adult ED that has hardly ever been investigated.

It is interesting that we found a female/male ratio of 1.6:1 in adolescents presenting with mental-health-related problems, whereas in the original overview, the overall sex distribution for adolescents presenting to our ED for any reason was almost balanced [1]. The finding that mental health conditions in female adolescents are more common is in line with several studies that found similar values of 62–65% females in their cohort [2,11]. However, this ratio may be distorted by repeated presentations, which are higher for both problems related to mental health and for females [12]. In 65 of the 612 included cases, patients had presented to our ED at least twice for psychiatric problems. Of these, 40 were only treated twice, but some individual patients were treated five to seven times, or in one case even 23 times. Data of previous studies show that repeated visits by the same patient usually account for a substantial portion of mental health-related ED-presentations (approximately 20–40%) [2,12,13,14].

Since the first onset of three quarters of mental health disorders is before the age of 18 years, it does not come as a surprise that 54% of our cohort had received psychiatric care prior to ED presentation [15].

Not only did we see fewer male adolescents in the ED, but they were also significantly less likely than females to have sought previous psychiatric assistance. The conclusion that more females than males seek psychiatric assistance is supported by a Swiss overview of psychiatric health care that focused on adolescents in their transitional years [7]. This may be the reason that males were significantly less likely to be admitted to our ED by healthcare professionals than were females.

The most frequent diagnoses among our cohort were reactions to severe stress and adjustment disorders, followed by disorders due to use of alcohol, intentional self-harm, and affective disorders. Nearly one fifth of the cohort presented with reactions to severe stress, and adjustment disorders which might be correlated with the pressure adolescents are under in their transitional life years. There is a difference to adulthood, since the most frequent mental health-related diagnosis in Swiss adults is depression, hence affective disorders [16]. Males were significantly more likely to present with disorders due to use of alcohol, which is in line with several Swiss observational studies [9,17,18]. A Swiss overview of psychiatric health in 2020 showed that the most frequent diagnoses in children and adolescents who were admitted to inpatient psychiatric care are depressive episodes (F32), followed by reactions to severe stress, and adjustment disorders (F43) [16]. Especially when it comes to acute alcohol intoxication, patients often stay at the ED for detoxication and go home as soon as they feel better, but they are rarely admitted to inpatient care.

Pre-existing disorders of adult personality and behaviour (F60–F69) were significantly less likely in males. However, it is controversial to make a diagnosis of these disorders at such a young age, since it must be based on a specific pattern of symptoms and behaviour throughout adolescence and adulthood. This emphasises the transitional age of adolescents, so that it is unusual to make a diagnosis at the age of 16 to 18 years. In this study, 17.3% of presentations of adolescents to the ED were for deliberate self-harm and males were significantly less likely to be affected. Lo et al. support this high prevalence, and even showed a threefold increase in intentional self-harm in American children aged 5 to 17 years over a study period of ten years (2007–2016) [3]. Male adolescents in our cohort less often reported previous self-harm, previous suicidal intentions, or previous suicide attempts. In addition, males had almost four times lower odds of presenting with previous suicide attempts compared to females, and this is similar to results of a Turkish study about non-suicidal self-harm, suicidal ideation and suicide attempts (SA) in children and adolescents (female/male ratio for SA = 4.83:1) [19]. It is crucial to identify and deal with non-suicidal self-harm and suicidal ideation, since they are known to be strong predictors for suicide. A study on suicide in England and Wales showed that 52% of children and adolescents who committed suicide had a history of self-harm [20].

Overall, approximately one fifth of our cohort reported attempted suicide prior to ED presentation. According to the WHO, self-harm is the leading cause of death in Swiss adolescents aged fifteen to nineteen years—so that it is essential to recognise self-harming behaviour [5]. This is of the greatest importance, as EDs are frequently the first point of contact with the health care system for many children and adolescents who self-harm. Therefore, we highlight the importance of establishing specific post-ED care for patients with self-harming behaviour.

The incidence of suicidal tendencies as the main reason for presentation to the ED (1.1%) was much less than the incidence of acute suicidal ideation in a specific assessment during their stay at the ED (31.7%). One explanation for the rather low percentage of acute suicidal tendency in our cohort might be a bias in triage categorisation, so that suicide attempts might not have been categorised as a primary mental health problem, but for example as trauma or as a neurological problem (e.g., self-poisoning). Otherwise, suicidal ideation is mostly associated with other problems, be they mental, social, or physical, and are therefore not necessarily the chief complaint. In addition, patients might not mention suicidal tendencies on their own due to shame or fear of stigmatisation. However, the high percentage of patients reporting acute suicidal ideation shows the importance of a thorough assessment in the emergency department, and this indicates that every patient presenting with mental health problems should be explicitly asked about suicidal thoughts and intentions.

Whereas young girls tended more to self-harming behaviour, 14.3% of males in our cohort stated that they represented a danger for fellow men and this was associated with threefold higher odds of external aggression in boys. Our data are therefore consistent with the view that females more often present with internalising diagnoses, whereas males have a higher rate of externalizing diagnoses [9]. This is in line with the aforementioned Turkish study which also showed the prominence of aggressive symptoms in males [19]. It is known that not only mental health issues and younger age per se, yet also male sex, ED admission in police custody as well as active substance use, are associated with a greater risk of violent events in the emergency department [21]. Possible parallels can be drawn, since our cohort showed that males were roughly two times more likely to be admitted by police. Furthermore, boys were significantly more likely to report consumption of alcohol and cannabis.

Not only were disorders due to use of alcohol the second most frequent reason for referral to the ED (17.6%), but overall, 43.3% of the cohort affirmed regular alcohol use. We were not able to draw a distinction between acute alcohol intoxication and harmful alcohol use or even alcohol dependence. Alcohol misuse was globally the highest risk factor for death among adolescents aged fifteen to nineteen years in 2013, similar to our results predominantly in males [22]. Furthermore, alcohol consumption poses a risk for suicide and is strongly associated with mental health disorders [4,8]. In addition to this, excessive alcohol consumption leads to an increase in violence and aggressiveness, trauma, and social marginalisation [17]. A Swiss report on addiction monitoring stated that adolescents mainly drink alcohol during the weekends, but when they drink they tend to consume large amounts [17,18]. This leads to binge drinking, particularly for adolescents or young adults aged 15 to 24 years.

More than a fifth (21.2%) of all cases reported smoking tobacco; however, from our data it was unclear whether the adolescents were smoking it daily or just on occasion. This percentage is in line with the prevalence of smoking tobacco in Swiss adolescents aged fifteen to nineteen years (21.1%) [18]. There was no significant difference in likelihood according to sex.

Overall, 26.3% of our cohort reported the use of cannabis, with males having double the odds for this. This percentage is similar to the lifetime prevalence of cannabis consumption in Swiss adolescents aged 15 to 19 years [18]. As the twelve month prevalence (20.8%) and the 30 day prevalence (9.4%) in the same subgroup are considerably lower, it suggests that many adolescents consume cannabis only on occasion, maybe even only to try it on a few opportunities. It would be interesting to see whether the twelve month and 30 day prevalence in our cohort were higher, but it was not possible to deduce this from our results. It may be relevant that adolescents who consume cannabis regularly, also tend to drink more alcohol, smoke regularly and use more other drugs [18].

3.3% to 4.7% of patients reported that they consumed either amphetamine, LSD, cocaine, or Ecstasy/MDMA. It is possible that the respective subgroups overlap, but this is only an assumption. Apart from LSD consumption, which was more likely for males than females, there was no significant sex difference to discern. The percentages in our cohort were noticeably higher than published values for the overall twelve month prevalence of drug use by Swiss adolescents (0.3% to 0.5%) [18]. This might be because our numbers represent the lifetime prevalence—which is supposedly higher. In general, it is presumed that the use of illegal drugs is underestimated in health surveys due to stigmatization and social undesirability, whereas this effect might be less marked during history taking by the ED doctor [18]. However, it would not come as a surprise that drug use among mentally ill adolescents is more likely than in the overall population, as this might be an expression of psychological distress, dysfunctional coping-strategies, or an attempt at self-healing.

A quarter of adolescents in our cohort were admitted to the ED by family members. A previous study also found that the family was the most common source of referral to the ED for children and adolescents, but with a higher percentage (49%) [23]. However, this study included children from the age of five years and it seems likely that the younger patients are, the more often they are admitted by family members. As mentioned before, males were significantly less likely to be admitted to the ED by health care professionals, suggesting that they are less likely to seek help earlier. They were, however, more likely to be escorted to the ED by police, in line with their higher likelihood of external aggressive behaviour.

According to a Swiss study, low educational attainment, migration background, loneliness, as well as unemployment mark risk factors for psychiatric diseases [16]. In total, 14% of adolescents in our study reported a migration background, with a significantly higher likelihood in males (OR 2.58, 1.61–4.11). A study in Vienna, Austria—a larger city but with a similar culture—found that 53.4% of minors presenting to the psychiatric emergency department had a migration background—which is clearly more than in our study [24]. They commented that emergency presentations from patients with migration background happen more often when the problems become overwhelming, rather than that they sought psychological help at an earlier point. Additionally, and contrary to our findings, even in patients with migration background, females presented significantly more often to the ED. Young adults with migration background presenting to the ED for mental health-related problems surely pose an interesting population and further characterisation is needed to understand this population more clearly.

A large portion of young adults mentioned familial problems, significantly less often in boys. Since adolescence is an important period for establishing social and emotional behaviour, conflicts might have crucial consequences [4]. In a Turkish study, relational problems were indeed the strongest predictor of suicide attempts in children [19]. The importance of a functional familial setting in adolescents and young adults is furthermore highlighted by Radde et al., who stated that dysfunctional communication with parents was a risk for suicidal tendencies [25]. In addition to familial problems, educational issues were frequent as well, although with a higher likelihood in males. Interestingly, studies showed that ED presentation in adolescents occurs more commonly during school months, which might be a result of increased stress levels [23].

More than half of our population were discharged home, which suggests that the clinical presentations in these patients were not as severe as to need further hospital or psychiatry treatment as an inpatient. The likelihood of males being able to return home was significantly higher—suggesting that disease patterns were more severe in girls. Additionally, males significantly more often presented with alcohol use disorders, but this less often resulted in inpatient admission.

This study has several strengths: It was conducted in an interdisciplinary ED and had broad inclusion criteria. Selection bias was minimised by careful screening and individual scrutiny of each report. In addition, there are hardly any previous data for this particular age-group focusing on mental-health-related ED visits. However, there are some limitations: Since the study was conducted as a retrospective analysis, there were no standardised questionnaires. Furthermore, for the statistical analysis we set ‘no information’ as being equivalent to a negation, as we assumed that only existing characteristics would be documented in the ED reports. However, the first assessment is usually performed by a general clinician rather than a psychiatrist, so there might be an information bias as not everyone inquired about all the parameters we analysed.

Additionally, data collection was based on diagnosis at triage, resulting not only in potential wrongful classification, yet also in possible underestimation of secondary diagnoses of mental health problems. Furthermore, data were gathered from one emergency department only, so data from similar clinics should be analysed too, in order to gather more information about adolescents with mental health-related problems. It should also be borne in mind that many adolescents hesitate to seek psychiatric assistance, so many cases are unreported.

As adolescents aged sixteen to eighteen years mark a transitional age group when it comes to health care services, we aimed to distinguish this specific subpopulation. Since presentation to our emergency department due to psychiatric problems was more frequent in younger than in older adults, and adolescence is known to be a vulnerable phase in which many mental health illnesses arise, it is important to understand this population [1]. Additionally, there is the problem in Switzerland that the responsibilities in health care for adolescents change with age. As mentioned before, in our health system child psychiatrists are responsible for patients up to their eighteenth birthday. In contrast, for emergency services, adolescents in our hospital already change to adult care at the age of sixteen, i.e., two years earlier. At our ED, adolescents are assessed by adult psychiatrists. If it is necessary for them to be admitted to inpatient care, however, they enter a child psychiatry unit. Therefore, it is of crucial importance that not only child psychiatrists, but also adult psychiatrists focus specifically on adolescents, as mental health illnesses tend to persist into adulthood. A change in care services at some point is inevitable. Additionally, since there is a different spectrum of psychiatric diseases in adolescents, explicit work up and collaboration of child and adult psychiatrists is of great importance.

## 5. Conclusions

Adolescents aged 16 to 18 years presenting to the ED with mental-health-related problems are an important subgroup of ED visitors. We should bear in mind that they are in a vulnerable phase of age and that there might be many different problems accompanying their psychiatric situation, e.g., familial or educational issues and substance use.

The high rate of patients reporting acute suicidal ideation on assessment shows how important it is to thoroughly evaluate this condition. Since patients often do not mention it spontaneously, it is crucial to assess adolescents explicitly for suicidal ideation. Furthermore, substance use as well as the familial and educational setting need to be investigated in adolescents presenting with mental-health-related problems.

Finally, since there might be a high rate of ED recidivism in patients with psychiatric problems and they tend to exhibit a higher risk of suicidal ideation, it is essential to provide adequate post-ED care.

To conclude, the study at hand shows the importance of mental health problems in young adults. Since adolescence is a vulnerable phase and disorders emerging in young adults continue to have substantial effects on life into adulthood, it is crucial to carefully attend to young adults’ needs when it comes to psychiatric care. However, to explore more variables related to mental health, further inquiries need to be conducted. Especially, it is important to perform assessments with standardised questionnaires to evaluate negative factors affecting mental health, as well as long-term follow-ups to see whether adequate post-ED care is established.

## Figures and Tables

**Figure 1 ijerph-19-13196-f001:**
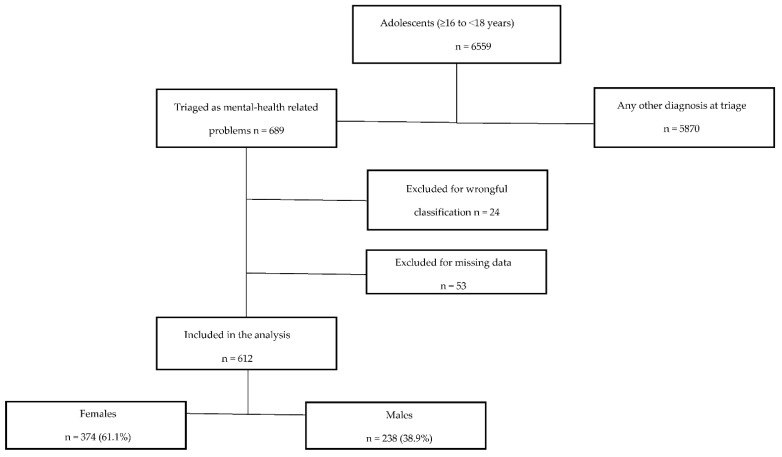
Flow chart of group selection.

**Figure 2 ijerph-19-13196-f002:**
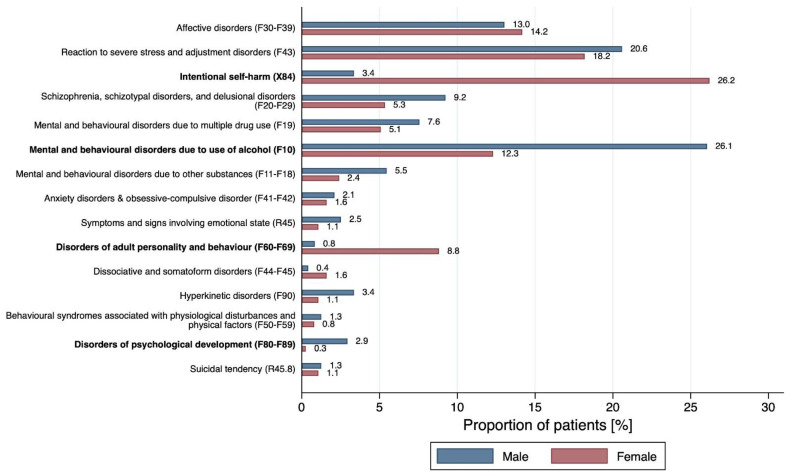
Sex differences of diagnosis groups according to ICD-10. Variables with significant differences between males and females are shown in bold (*p* < 0.05).

**Figure 3 ijerph-19-13196-f003:**
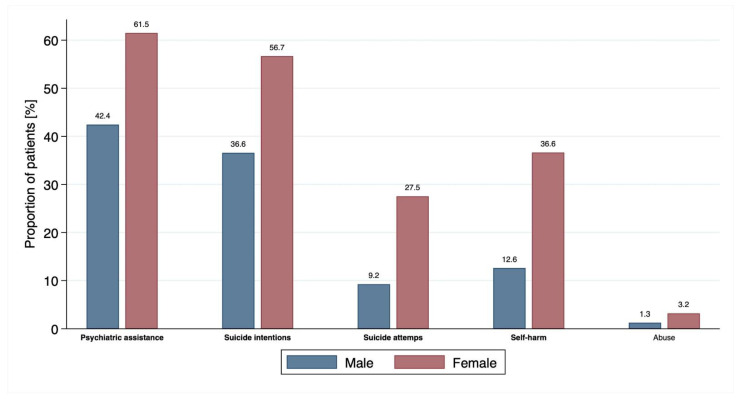
Sex difference of history of psychiatric problems. Variables with significant differences (*p* < 0.05) between males and females are shown in bold.

**Table 1 ijerph-19-13196-t001:** Diagnosis groups according to ICD-10.

	Male n (%)	Female n (%)	Total (%)	OR (95%CI)	*p*
Reaction to severe stress, and adjustment disorders (F43)	49 (20.6)	68 (18.2)	117 (19.1)	1.17 (0.77–1.76)	0.461
Mental and behavioural disorders due to use of alcohol (F10)	62 (26.1)	46 (12.3)	108 (17.6)	2.51 (1.65–3.83)	<0.001
Intentional self-harm (X84)	8 (3.4)	98 (26.2)	106 (17.3)	0.10 (0.05–0.21)	<0.001
Affective disorders (F30–F39)	31 (13.0)	53 (14.2)	84 (13.7)	0.91 (0.56–1.46)	0.688
Schizophrenia, schizotypal and delusional disorders (F20–F29)	22 (9.2)	20 (5.3)	42 (6.9)	1.80 (0.96–3.38)	0.066
Mental and behavioural disorders due to multiple drug use (F19)	18 (7.6)	19 (5.1)	37 (6.0)	1.53 (0.79–2.98)	0.212
Disorders of adult personality and behaviour (F60–F69)	2 (0.8)	33 (8.8)	35 (5.7)	0.09 (0.02–0.37)	<0.001
Mental and behavioural disorders due to other psychoactive substances (F11–F18)	13 (5.5)	9 (2.4)	22 (3.6)	2.34 (0.99–5.57)	0.054
Hyperkinetic disorders (F90)	8 (3.4)	4 (1.1)	12 (2.0)	3.22 (0.96–10.81)	0.059
Anxiety disorders, and obsessive compulsive disorders (F41–F42)	5 (2.1)	6 (1.6)	11 (1.8)	1.32 (0.40–4.36)	0.653
Symptoms and signs involving emotional state (R45)	6 (2.5)	4 (1.1)	10 (1.6)	2.39 (0.67–8.57)	0.18
Disorders of psychological development (F80–F89)	7 (2.9)	1 (0.3)	8 (1.3)	11.30 (1.38–92.46)	0.024
Dissociative and somatoform disorders (F44–F45)	1 (0.4)	6 (1.6)	7 (1.1)	0.26 (0.03–2.16)	0.212
Suicidal tendency (R45.8)	3 (1.3)	4 (1.1)	7 (1.1)	1.18 (0.26–5.32)	0.829
Behavioural syndromes associated with physiological disturbances and physical factors (F50–F59)	3 (1.3)	3 (0.8)	6 (1.0)	1.58 (032–7.89)	0.578

**Table 2 ijerph-19-13196-t002:** Acute suicidal ideation and history of psychiatric problems.

	Male n (%)	Female n (%)	Total (%)	OR (95%CI)	*p*
Acute suicidal ideation	59 (24.8)	135 (36.1)	194 (31.7)	0.58 (0.41–0.84)	0.004
Acute danger for others	24 (14.3)	19 (5.1)	53 (8.7)	3.11 (1.73–5.60)	<0.001
History of psychiatric problems					
Previous psychiatric assistance	101 (42.4)	230 (61.5)	331 (54.1)	0.46 (0.33–0.64)	<0.001
Suicidal intentions	87 (36.6)	212 (56.7)	299 (48.9)	0.44 (0.32–0.61)	<0.001
Suicide attempt	22 (9.2)	103 (27.5)	125 (20.4)	0.27 (0.16–0.44)	<0.001
Self-harm	30 (12.6)	137 (36.6)	167 (27.3)	0.25 (0.16–0.39)	<0.001
Abuse	3 (1.3)	12 (3.2)	15 (2.5)	0.39 (0.11–1.38)	0.143

**Table 3 ijerph-19-13196-t003:** History of pre-existing psychiatric diagnoses.

History of Psychiatric Diagnoses	Male n (%)	Female n (%)	Total (%)	OR (95%CI)	*p*
Disorders of adult personality and behaviour (F60–F69)	6 (2.5)	50 (13.4)	56 (9.2)	0.17 (0.07–0.40)	<0.001
Affective disorders (F30–F39)	7 (2.9)	40 (10.7)	47 (7.7)	0.25 (0.11–0.57)	0.001
Hyperkinetic and conduct disorders (F90–F91)	24 (10.1)	19 (5.1)	43 (7.0)	2.10 (1.12–3.92)	0.020
Mental and behavioural disorders due to substance abuse (F10–F19)	14 (5.9)	17 (4.5)	31 (5.1)	1.31 (0.63–2.71)	0.463
Reaction to severe stress, and adjustment disorders (F43)	7 (2.9)	16 (4.3)	23 (3.8)	0.68 (0.27–1.67)	0.399
Eating disorders (F50)	0 (0.0)	16 (4.3)	16 (2.6)	-	-
Schizophrenia, schizotypal and delusional disorders (F20–F29)	7 (2.9)	6 (1.6)	13 (2.1)	1.86 (0.62–5.60)	0.271
Disorders of psychological development (F80–F89)	5 (2.1)	6 (1.6)	11 (1.8)	1.32 (0.40–4.36)	0.653
Anxiety disorders, and obsessive compulsive disorders (F41–F42)	1 (0.4)	4 (1.1)	5 (0.8)	0.39 (0.04–3.51)	0.401
Dissociative and somatoform disorders (F44–F45)	1 (0.4)	4 (1.1)	5 (0.8)	0.39 (0.43–3.51)	0.401
Mental retardation (F70–F79)	4 (1.7)	1 (0.3)	5 (0.8)	6.38 (0.71–57.39)	0.098

**Table 4 ijerph-19-13196-t004:** Source for admission.

Source for Admission	Male n (%)	Female n (%)	Total (%)	OR (95%CI)	*p*
Family	65 (27.3)	89 (23.8)	154 (25.2)	1.20 (0.83–1.74)	0.329
Peers	41 (17.2)	53 (14.2)	94 (15.4)	1.26 (0.81–1.97)	0.307
Health care professionals	20 (8.4)	70 (18.7)	90 (14.7)	0.40 (0.24–0.67)	0.001
Adviser	27 (11.3)	43 (11.5)	70 (11.4)	0.99 (0.59–1.64)	0.954
Police	26 (10.9)	22 (5.9)	48 (7.8)	1.96 (1.08–3.55)	0.026
Self-admission	18 (7.6)	27 (7.2)	45 (7.4)	1.05 (0.57–1.95)	0.874
No information	41 (17.2)	70 (18.7)	111 (18.1)	0.90 (0.59–1.38)	0.641

**Table 5 ijerph-19-13196-t005:** Substance use.

Substance Use	Male n (%)	Female n (%)	Total (%)	OR (95%CI)	*p*
Alcohol	129 (54.2)	136 (36.4)	265 (43.3)	2.07 (1.49–2.88)	<0.001
Cannabis	83 (34.9)	78 (20.9)	161 (26.3)	2.03 (1.41–2.93)	<0.001
Amphetamine	9 (3.8)	11 (2.9)	20 (3.3)	1.30 (0.53–3.18)	0.57
LSD	14 (5.9)	7 (1.9)	21 (3.4)	3.28 (1.30–8.24)	0.012
Cocaine	11 (4.6)	18 (4.8)	29 (4.7)	0.96 (0.44–2.07)	0.914
Ecstasy/MDMA	11 (4.6)	15 (4.0)	26 (4.2)	1.16 (0.52–2.57)	0.715
Methamphetamine/Crystal Meth	3 (1.3)	1 (0.3)	4 (0.7)	4.76 (0.49–46.05)	0.178
Ketamine	1 (0.4)	1 (0.3)	2 (0.3)	1.57 (0.10–25.28)	0.749
Heroin	4 (1.7)	6 (1.6)	10 (1.6)	1.05 (0.29–3.75)	0.942
Psilocybin	1 (0.4)	2 (0.5)	3 (0.5)	0.78 (0.07–8.70)	0.844
Smoking tobacco	54 (22.7)	76 (20.3)	130 (21.2)	1.15 (0.78–1.71)	0.485

**Table 6 ijerph-19-13196-t006:** Social impact.

Social Impact	Male n (%)	Female n (%)	Total (%)	OR (95%CI)	*p*
Familial problems	54 (22.7)	117 (31.3)	171 (27.9)	0.64 (0.44–0.94)	0.021
Educational problems	38 (16.0)	38 (10.2)	76 (12.4)	1.68 (1.04–2.72)	0.035
Problems with peers	9 (3.8)	14 (3.7)	23 (3.8)	1.01 (0.43–2.37)	0.981
Death/disease of a family member	12 (5.0)	30 (8.0)	42 (6.9)	0.61 (0.31–1.21)	0.159
Witness to trauma of another person	1 (0.4)	1 (0.3)	2 (0.3)	1.58 (0.10–25.28)	0.749
Migration background	50 (21.0)	35 (9.4)	85 (13.9)	2.58 (1.61–4.11)	<0.001

**Table 7 ijerph-19-13196-t007:** Route of discharge.

Route of Discharge	Male n (%)	Female n (%)	Total (%)	OR (95%CI)	*p*
Home	135 (56.7)	178 (47.6)	313 (51.1)	1.44 (1.04–2.00)	0.028
Extern hospital	87 (36.6)	164(43.9)	251 (41.0)	0.74 (0.53–1.03)	0.074
Hospital admission	9 (3.8)	26 (7.0)	35 (5.7)	0.53 (0.24–1.14)	0.105
No information	7 (2.9)	6 (1.6)	13 (2.1)	1.86 (0.62–5.60)	0.271

## Data Availability

Not applicable.

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
