# Peer review of "Presentations to the Emergency Department for Problems Related to Mental Health: Sex Differences in Adolescents"

_ijerph, 2022, doi:10.3390/ijerph192013196_

Round 1

Reviewer 1 Report

This is an increasingly worrying topic of interest in the developed world. Mental health problems are increasing alarmingly, especially in the adolescent population. It would have been interesting to collect more information, which would allow exploring more variables related to mental health disorders. This work adds little information beyond exploring gender differences. It would be interesting to include variables about the family typology, socioeconomic level, family history so that this descriptive study would allow hypotheses and future lines of research to be proposed.

1. The introduction is well thought out but the objective is not clear.

2. Specify the inclusion and exclusion criteria.

3. The results are well exposed, but they are not very relevant due to the scarcity of variables collected.

4. The breakdown by subsections according to results in the discussion does not seem very appropriate. It would be better exposed without subsections.

5. No prospective research or applicability in clinical practice is appreciated.

Author Response

Dear Editor, dear Reviewer 1

We appreciate your interest and helpful attentative peer-review of this paper:

Presentation to the Emergency Department for Problems Related to Mental Health: Sex Differences in Adolescents

We were pleased to hear that IJERPH is interested in a revised version of our manuscript. We are grateful to the reviewers for their helpful suggestions and comments. All comments have been addressed in the revised version. In more detail:

Review 1

This is an increasingly worrying topic of interest in the developed world. Mental health problems are increasing alarmingly, especially in the adolescent population. It would have been interesting to collect more information, which would allow exploring more variables related to mental health disorders. This work adds little information beyond exploring gender differences. It would be interesting to include variables about the family typology, socioeconomic level, family history so that this descriptive study would allow hypotheses and future lines of research to be proposed.

  1. The introduction is well thought out but the objective is not clear.

Reply: We thank the reviewer for this important note. The main goal of the study (characterizing young adults presenting to the ED for psychiatric problems) is now mentioned in the first paragraph of the introduction to clarify the objective at the beginning. Furthermore, the objective is highlighted more clearly in the last paragraph of the introduction.

  1. Specify the inclusion and exclusion criteria.

Reply: Done as recommended.

  1. The results are well exposed, but they are not very relevant due to the scarcity of variables collected.

Reply: A very important comment, however, since we only extracted data from medical reports of the ED visits, it was not possible to elaborate more detailed data. Since few data exist on this specific age group presenting to the ED with mental health related problems, our goal with the study at hand was to make a characterization of these adolescents. The study may lay the foundation for further inquiries, which are certainly important to gather an even better understanding of young adults with psychiatric problems. Further studies need to be done on this topic. For this reason, we highlighted the importance of further investigation in the introduction as well as the conclusion.

  1. The breakdown by subsections according to results in the discussion does not seem very appropriate. It would be better exposed without subsections.

Reply: Thank you very much for the suggestion, we have taken out the headings. I hope that it will still be good to read and understand without them.

  1. No prospective research or applicability in clinical practice is appreciated.

Reply: As recommended, we added suggestions for further research in the conclusion.

Thank you for the thorough and helpful reviewing. We certainly hope that our paper is soon to be accepted and published in the International Journal of Environmental Research and Public Health

We wish you best regards and good health.

Raffaela Flury

Jolanta Klukowska-Rötzler

Reviewer 2 Report

I think this is an important retrospective study of a vulnerable population. Given the conclusions of the study regarding the need to thoroughly assess these adolescents, I'm left wanting more regarding post-ED care versus post-ED discharge.  

I hope that your research inspires others to take a next step and research aftercare, which was beyond the scope of your study. 

Line 367 speaks to 21.2% percent of the youth smoking.  I would recommend operationalizing "smoking".  I took it to mean tobacco but that is an assumption.

Line 445 to 448  Potential recommendation: to elaborate more on the importance of the study given that adult psychiatrists are doing the evaluation.  Line 448 states, "It is still unclear whether this difference is ultimately of importance" This may be my US training, but I think these are the folks that really need the information you are providing about adolescent presentation in the ED.  My apologies, if I miss understand, but I'm thinking adult psychiatrists that do the evaluations are a potential target audience for your study.

Author Response

Dear Editor, dear Reviewer 2

We appreciate your interest and helpful attentative peer-review of this paper:

Presentation to the Emergency Department for Problems Related to Mental Health: Sex Differences in Adolescents

We were pleased to hear that IJERPH is interested in a revised version of our manuscript. We are grateful to the reviewers for their helpful suggestions and comments. All comments have been addressed in the revised version. In more detail:

Review 2

I think this is an important retrospective study of a vulnerable population. Given the conclusions of the study regarding the need to thoroughly assess these adolescents, I'm left wanting more regarding post-ED care versus post-ED discharge.  

I hope that your research inspires others to take a next step and research aftercare, which was beyond the scope of your study. 

Line 367 speaks to 21.2% percent of the youth smoking.  I would recommend operationalizing "smoking".  I took it to mean tobacco but that is an assumption.

Reply: As the reviewer assumed correctly, we are talking of smoking tobacco. We now specified it accordingly.

Line 445 to 448  Potential recommendation: to elaborate more on the importance of the study given that adult psychiatrists are doing the evaluation.  Line 448 states, "It is still unclear whether this difference is ultimately of importance" This may be my US training, but I think these are the folks that really need the information you are providing about adolescent presentation in the ED.  My apologies, if I miss understand, but I'm thinking adult psychiatrists that do the evaluations are a potential target audience for your study.

Reply: We thank the reviewer for this important recommendation. We made changes accordingly, to highlight the importance of child AND adult psychiatrists in mental health care for adolescents.

We certainly hope that our paper is soon to be accepted and published in the International Journal of Environmental Research and Public Health

We wish you best regards and good health.

Raffaela Flury, Jolanta Klukowska-Rötzler

Reviewer 3 Report

The article presented for review is an interesting report on the analysis of medical data on mental problems of adolescents aged 16 – 18 years presenting to the department (ED) at the University Hospital in Bern (Inselspital), Switzerland. The issues raised should be considered important. The results of the conducted analyses are presented in a clear and legible manner and allow for a preliminary estimation of the scale of adolescent mental problems, the nature of these problems and gender differences in this area. In my opinion, the text is well prepared. It would be interesting to relate the results of the analyses to data from other ED in Switzerland or other countries, which would allow to determine whether the nature and severity of adolescent mental health problems and the recorded sex differences correspond only to a specific group covered by the analyses or to adolescents 16 – 18 in general. I only have a formal remark about placing the titles of the tables, which should be above the tables

Author Response

Dear Editor, dear Reviewer 3

We appreciate your interest and helpful attentative peer-review of this paper:

Presentation to the Emergency Department for Problems Related to Mental Health: Sex Differences in Adolescents

We were pleased to hear that IJERPH is interested in a revised version of our manuscript. We are grateful to the reviewers for their helpful suggestions and comments. All comments have been addressed in the revised version. In more detail:

Review 3

The article presented for review is an interesting report on the analysis of medical data on mental problems of adolescents aged 16 – 18 years presenting to the department (ED) at the University Hospital in Bern (Inselspital), Switzerland. The issues raised should be considered important. The results of the conducted analyses are presented in a clear and legible manner and allow for a preliminary estimation of the scale of adolescent mental problems, the nature of these problems and gender differences in this area. In my opinion, the text is well prepared. It would be interesting to relate the results of the analyses to data from other ED in Switzerland or other countries, which would allow to determine whether the nature and severity of adolescent mental health problems and the recorded sex differences correspond only to a specific group covered by the analyses or to adolescents 16 – 18 in general. I only have a formal remark about placing the titles of the tables, which should be above the tables

Reply: We were very pleased with the reviewer’s positive feedback! As well as the reviewer, we would also be very interested in comparing our data to other Emergency Departments in Switzerland and other countries. We hope that our study, which to our knowledge is the first on this subject, will lay the foundation for following inquiries in the mental well-being of young adults.

As for placement of the titles of the tables, we changed that as recommended.

We certainly hope that our paper is soon to be accepted and published in the International Journal of Environmental Research and Public Health

We wish you best regards and good health.

Raffaela Flury

Jolanta Klukowska-Rötzler
